# Transcriptome and Metabolome Analyses Offer New Insights into Bolting Time Regulation in Broccoli

**DOI:** 10.3390/ijms26083726

**Published:** 2025-04-15

**Authors:** Linqian Kuang, Yue Zhang, Nan Zhang, Yangyong Zhang, Honghao Lv, Yong Wang, Mu Zhuang, Limei Yang, Ke Huang, Zhansheng Li, Jialei Ji

**Affiliations:** 1Key Laboratory for Vegetable Biology of Hunan Province, Engineering Research Center for Horticultural Crop Germplasm Creation and New Variety Breeding, Ministry of Education, Hunan Agricultural University, Changsha 410128, China; kuanglinqian@163.com (L.K.); huangkeqy@hotmail.com (K.H.); 2State Key Laboratory of Vegetable Biobreeding, Institute of Vegetables and Flowers, Chinese Academy of Agricultural Sciences, Beijing 100081, China; zhangyue199901@126.com (Y.Z.); zhangnan620421@163.com (N.Z.); zhangyangyong@caas.cn (Y.Z.); lvhonghao@caas.cn (H.L.); wangyong@caas.cn (Y.W.); zhuangmu@caas.cn (M.Z.); yanglimei@caas.cn (L.Y.)

**Keywords:** broccoli, bolting time, transcriptome, metabolome, floral pathway, sugar signal, hormone signaling

## Abstract

The globular buds and stems are the main edible organs of broccoli. Bolting is an important agronomic trait, and the timing of its occurrence is particularly critical when breeding and domesticating broccoli. The molecular mechanism that regulates broccoli bolting time is not well-understood. In this study, the apical flower bud and leaf tissues of two broccoli varieties with different bolting intensities were selected for metabolome and transcriptome analyses. In the apical flower buds of early-bolting B2554 and late-bolting B2557, 1094 differentially expressed genes and 206 differentially accumulated metabolites were identified. In the leaves, 487 differentially expressed genes and 40 differentially accumulated metabolites were identified. In the floral pathway, the expression of *FLC* (*FLOWERING LOCUS C*) was significantly upregulated, and that of *FT* (*FLOWERING LOCUS T*) was significantly downregulated in the late-bolting plants, indicating their possible role in suppressing bolting. In addition, significant differences were identified in the sucrose synthesis and transport, hormone synthesis, and signal transduction processes in early-bolting B2554 and late-bolting B2557. Sucrose accumulation in the leaves and apical flower buds of the early-bolting plants was about 1.3 times higher than in the late-bolting plants. Indole-3-acetic acid (IAA) and abscisic acid (ABA) accumulation in the apical flower buds of the late-bolting plants was more than twice that in the early-bolting plants. Jasmonic acid (JA) accumulation in the apical flower buds of the late-bolting plants was more than ten times higher than in the early-bolting plants. Phenolic acids may affect the bolting time of broccoli. This study offers new insights into the regulation mechanism of broccoli bolting and provides some potential molecular targets to include in breeding methods that regulate bolting time.

## 1. Introduction

Broccoli (*Brassica oleracea* L. var. *italica* Plenck) is a vegetable crop belonging to the Brassica genus and Brassicaceae family and is known as the “crown of vegetables” due to its rich nutritional composition. The historical origin of broccoli can be traced back to Italy, where it became popular worldwide in the 19th century and has a very long history of cultivation [1,2]. The globular flower buds and stems are the main edible organs of broccoli. The development of the apical flower buds affects the formation of flower balls and flowering. In this process, the apical meristem of the nutritious stem transforms into the flower-producing meristem to enter the reproductive stage [3]. The transition from vegetative growth to reproductive growth at the right time is crucial for propagating broccoli offspring, especially when breeding and domesticating broccoli. Premature bolting leads to the early formation of flower buds, which prevents large flower balls from forming and affects the normal development of flower balls. This not only reduces the broccoli yield. but also significantly impacts its quality. Therefore, a correctly timed bolting stage has considerable economic value.

Bolting and flowering are key developmental stages in the plant’s life cycle and are strictly controlled by many complex regulatory networks. These regulatory networks involve responding to numerous environmental and internal signals that alter the floral development genes. These genes quantitatively regulate the development of the stem’s apical meristem, thereby determining whether flowering occurs and its precise timing [4]. Therefore, the control of flowering time (Ft) is plastic. In *Arabidopsis thaliana*, six pathways regulating flowering have been identified including the photoperiod pathway, the vernalization pathway, the ambient temperature pathway, the autonomous pathway, the gibberellin pathway, and the aging pathway. Different pathways control the flowering time by responding to endogenous signals and changes in the external environment [5]. In addition to these factors, endogenous plant hormones are another chemical component that has always influenced flowering [6]. IAA, cytokinin (CTK), gibberellin (GA), ABA, and ethylene (ETH) are the five important hormones in plants, and these five hormones and their related metabolites have been confirmed to participate in the regulation of the initial flowering period in plants [7]. Disrupting the biosynthesis, polar transport, and signaling pathways of IAA leads to delayed flowering or no flowering at all [8]. GA inhibits the expression of *FLC* by promoting the degradation of the DELLA protein, thereby advancing the flowering period of *A. thaliana* [9,10]. ABA and ETH are generally thought to delay the flowering time by regulating the accumulation of the DELLA protein [11,12]. In addition, JA and salicylic acid (SA) finely regulate the flowering time by interacting with these hormones. Plant hormones generally do not function in a single way but are more often involved in complex signaling networks [13,14]. Notably, recent advances have revealed that carbohydrates are involved in these known flowering pathways, suggesting that they play a synergistic role in regulating flowering time [15,16]. Carbohydrates, such as sucrose, accumulate in the plant’s phloem and stem tip tissues to promote stem growth and transformation. Studies have shown that the level of sucrose in *A. thaliana* rises rapidly during flower induction, but its flowering time is affected by different sucrose concentrations. A low concentration of sucrose promotes flowering, while a high concentration inhibits it [17]. Sugar promotes or inhibits flowering, depending on the transmutase activity and the concentration of trehalose-6-phosphate (T6P) [18,19]. Sucrose signaling molecules can be distributed from other tissues to the buds and flowers via sucrose (Suc) transporter 9 (SUC9), which inhibits miR156 expression and induces flowering [20,21]. The hydrolysis products of sucrose, glucose, and fructose as well as the downstream metabolite trehalose-6-phosphate also act as signaling molecules affecting flowering [22]. In recent years, transcriptomic approaches have been widely employed to investigate the molecular mechanisms regulating the flowering time in crops, with key regulatory components identified through differential gene expression analysis. For instance, the transcriptome analysis of early-flowering (C491) and late-flowering (B602) cabbage (*Brassica oleracea*) inbred lines revealed that the MADS-box genes *BoSEP2-1* and *BoSEP2-2* exhibited significantly reduced expression levels in late-flowering materials and harbored non-synonymous SNP variants. These findings suggest their potential role in regulating cabbage flowering time [23]. In other Brassicaceae crops, transcriptomic approaches have been instrumental in dissecting conserved and species-specific regulatory modules of flowering time, as demonstrated in Chinese cabbage (*Brassica rapa*) and radish (*Raphanus sativus*) [24,25,26].

In this context, the aim of this study was to understand the abundance profiles of genes and metabolites related to flowering time regulation in different pathways involved in flowering in broccoli during the bolting period. Therefore, two broccoli varieties with different bolting intensities were selected for the study. B2554 is an early-bolting self-crossing broccoli variety, and B2557 is a late-bolting self-crossing variety. We focused on the molecular mechanisms and metabolic differences between these plant varieties when their stem apical inflorescence meristem was transitioning to the flower bud of broccoli, thus increasing our understanding of how bolting intensity is genetically controlled in the different varieties.

## 2. Results

### 2.1. Overview of the Metabolites Involved in the Development of Early- and Late-Bolting Broccoli

In order to investigate the dynamic changes of compounds involved in the bolting and budding processes of broccoli, the primary and secondary metabolites in the apical flower bud meristems and leaves were identified and analyzed using UPLC-MS/MS technology. A total of 1403 metabolites were identified (Appendix A). The identified metabolites could be subdivided into 12 categories: 245 amino acids and derivatives (17.46%), 185 phenolic acids (13.19%), 152 lipids (10.83%), 145 alkaloids (10.33%), 141 flavonoids (10.05%), 85 organic acids (6.06%), 83 lignins and coumarins (5.92%), 74 nucleotides and derivatives (5.27%), 37 terpenoids (2.64%), 19 quinones (1.35%), 2 tannins (0.15%), and 235 other metabolites (16.75%; Figure 1A). Based on the cluster analysis of metabolite abundances, the 24 samples were divided into 8 different clusters, with the detected metabolites in the apical flower buds and leaves clearly distinguished (Figure 1B). Meanwhile, the PCA results clearly differentiated between the metabolic profiles of the apical flower buds and leaves. The first and second principal components explained 39.92% and 11.44% of the variation, respectively, and the samples from the same varieties and growth stages were repeatedly clustered together, which confirmed the reliability of the sampling method (Figure 1C).

Differentially accumulated metabolites (DAMs) were analyzed using a variable prediction significance (VIP) threshold > 1, and a *t*-test *p*-value < 0.05. The differential expression analysis of the apical flower bud tissues from B2554 and B2557 revealed 369 DAMs during the first growth period (1B7_vs_1B4) and 458 DAMs during the second growth period (2B7_vs_2B4). The differential expression analysis of the leaf tissues revealed 125 DAMs during the first growth period (1L7_vs_1L4) and 362 DAMs during the second growth period (2L7_vs_2L4). The upregulated and downregulated differential metabolites are shown in Figure 1D.

There were 206 DAMs in common between 1B7_vs_1B4 and 2B7_vs_2B4 and 40 DAMs in common between 1L7_vs_1L4 and 2L7_vs_2L4, and six DAMs were found in all four pairwise comparisons. In addition, there were 50 DAMs in common between 1B7_vs_1B4 and 1L7_vs_1L4 and 149 DAMs in common between 2B7_vs_2B4 and 2L7_vs_2L4 (Figure 1E). We analyzed the common differentially accumulated metabolites detected above. In general, phenolic acids were the most different metabolites in the apical flower buds of two broccoli plants with different bolting intensity, while flavonoids were the most different metabolites in the leaves. Then, in the comparison between the apical flower buds and the leaves in the first period, the most different metabolites were flavonoids, while in the second period, the most different metabolites were phenolic acids (Appendix A).

### 2.2. Transcriptome Analysis of the Development of Early- and Late-Bolting Broccoli

In total, the transcriptomes of 24 tissue samples were analyzed. After removing the low-quality raw sequenced reads, approximately 176.79 Gb of clean data was obtained. Each sample had nearly 6 Gb of clean data, with an average GC content of 47.57%. The Q30 range varied from 92.02% to 95.11%, with an average of 93.76%. Therefore, the sequencing quality was generally good. After quality control, the clean reads were mapped to the specified reference genome. Finally, 49,724 single genes were assembled from an average of 46.47 million mapped reads per library, with a mapping rate of 94.56% for all libraries (Appendix A). The PCA analysis showed that the first and second principal components explained 43.85% and 7.75% of the variation, respectively (Figure 2A). There was a high similarity among the three biological replicates (Figure 2B). These results also indicated that the sequencing results were reliable.

The FPKM values of each sample’s library were collected and analyzed to detect the gene expression profile. The differential expression analysis between the apical flower buds of early-bolting B2554 and late-bolting B2557 revealed 3559 differentially expressed genes (DEGs) in the first growth period and 2657 DEGs in the second growth period. The differential expression analysis of the leaves revealed 1202 DEGs in the first growth period and 4529 DEGs in the second growth period. Finally, we identified 180 DEGs in common in the four comparison groups (Figure 2C). These DEGs were classified into upregulated and downregulated transcripts based on their gene expression patterns. It is worth noting that the number of up- and downregulated DEGs in the apical flower buds from the first growth period was greater than from the second period (Figure 2D).

In this study, the enrichment analysis was conducted using Gene Ontology (GO) and the Kyoto Encyclopedia of Genes and Genomes (KEGG) to analyze the DEGs involved in broccoli bolting and understand their biological functions. The GO annotation highlights the enrichment of DEGs in various processes. In the comparison groups of apical flower bud and leaf tissue of the first growth period, the common GO term included “auxin biosynthetic process” (Appendix A). The KEGG enrichment analysis results for each comparison group were also summarized. In 1B7_vs_1B4 and 2B7_vs_2B4, the common KEGG pathways significantly enriched included “biosynthesis of secondary metabolites”, “metabolic pathways”, “tryptophan metabolism”, “plant hormone signal transduction”, and “flavonoid biosynthesis”. Moreover, “photosynthesis-antenna proteins” and “photosynthesis” were the most significantly enriched in 2B7_vs_2B4. In contrast, “metabolic pathways” and “tryptophan metabolism” pathways in the 1L7_vs_1L4 and 2L7_vs_2L4 comparison groups were significantly enriched. In addition, during the first growth period, 1B7_vs_1B4 and 1L7_vs_1L4 included “tryptophan metabolism”, “biosynthesis of secondary metabolites”, and “metabolic pathways”. During the second growth period, 2B7_vs_2B4 and 2L7_vs_2L4 included “tryptophan metabolism”, “metabolic pathways”, “fatty acid elongation”, and “plant hormone signal transduction” (Appendix A). These results suggest that the biosynthesis of secondary metabolites, metabolism, and plant hormone signal transduction may play an important role in the bolting process in broccoli.

### 2.3. Validation and Analysis by qRT-PCR

We speculated that the emergence of flower buds after bolting may be affected by some genes in the flowering pathway, so we selected nine genes involved in the flowering pathway to verify their expression patterns. The relative expression levels of these genes in eight tissues from the two broccoli varieties were similar to the expression trends observed in their transcriptomes (Figure 3). A comparative analysis of the expression patterns of the *BoLFY* (*BolC2t12224H*), *BoSEP2* (*BolC5t35090H*), and *BoSEP3* (*BolC5t31318H*) genes in the apical flower buds revealed that their expression patterns in the apical flower buds may be closely associated with bolting time. In addition, it can be reasonably inferred that the differences in the expression patterns of *BoFLC*, *BoFT*, *BoTOE1*, *BoTFL1*, and *BoTAA1* between the early- and late-bolting broccoli varieties may be potential signals regulating the bolting time.

### 2.4. Differential Expression in the Flowering Pathway

In this study, we analyzed the gene expression patterns associated with the six major pathways that regulate flowering. In light of the lack of data on how the bolting time in broccoli is genetically controlled, we took the flowering time (Ft) genes from *A. thaliana* and used them as a reference to search for homologs in broccoli. The analysis identified 292 Ft gene homologs in broccoli using the 155 flowering time-related genes identified in *A. thaliana* (Appendix A). These genes were divided into six categories based on the flowering regulatory pathways and included C/L/P (circadian clock pathway, light signaling pathway, and photoperiod pathway), V (vernalization pathway), A (autonomous pathway), G/M (gibberellin signaling and metabolism), D/M (development and metabolism response), I (integrator), and Age (age pathway) [26]. We found 39 DEGs associated with the Ft homologs in the early- and late-bolting plants, and most of these genes were found to be significantly different in the apical flower buds (Figure 4). Among these 39 genes, 12 were associated with C/L/P (30.77%), 9 were associated with D/M (23.08%), 4 were associated with V (10.26%), 7 were associated with G/M (17.95%), 3 were associated with Age (7.69%), and 4 were associated with I (10.26%).

The *FT* gene encodes mobile florigen, which is mainly expressed in the leaves and then transported through the phloem to the apical meristem to induce flowering [27]. We observed that the expression of *BoFT* in the leaves was higher than in the apical meristem. Although the expression difference of *BoFT* in 1L7_vs_1L4 was not significant, it was significantly different in 2L7_vs_2L4, with higher expression in the leaves of early-bolting B2554. Meanwhile, the expression pattern of *BoFT* in 2B7_vs_2B4 was consistent with that in 2L7_vs_2L4, which was also in line with the phenotypes we observed. Three homologous genes of *FD* (*FLOWERING LOCUS D*) were retrieved in broccoli. The expression of the three *BoFDs* was higher in the apical flower buds. Their expression levels in 1B7_vs_1B4 and 2B7_vs_2B4 were significantly different. Moreover, their expression was higher in the early-bolting B2554 during the first growth stage and in the late-bolting B2557 during the second growth stage. Compared with 1B7 and 2B7, the expression of *BoSOC1*, a gene located downstream of *FT*, was higher in 1B4 and 2B4, and the difference was also significant in 1B7_vs_1B4. In addition, the expression patterns of the downstream stem tip meristem genes *BoLFY*, *BoAP1*, and *BoFUL* were relatively similar in the apical flower buds, with all three showing significantly different expression patterns in 1B7_vs_1B4. *BoSEPs* were significantly different in 1B7_vs_1B4 and 2B7_vs_2B4. All of these genes showed higher expression in early-bolting B2554, thereby agreeing with the phenotype we observed. The expression of *BoAGL24* was significantly different in 1B7_vs_1B4, 2B7_vs_2B4, and 2L7_vs_2L4 and was higher in early-bolting B2554. In addition, the expression of the repressor *TFL1* (*TERMINAL FLOWER 1*) also supported their role in the observed phenotypes. *BoTFL1* exhibited significant differences in expression levels between 1B7_vs_1B4 and 2B7_vs_2B4, with higher expression in late-bolting B2557.

The expression levels of certain upstream inhibitors of *BoFT*, *BoTOE1-1*, and *BoTOE1-2* were significantly different in the apical flower buds between the two growth stages, with higher expression levels in late-bolting B2557. In contrast, the expression level of *BoTOE2* was significantly higher in the apical flower buds of early-bolting B2554. *BoTEM1*, which inhibits *FT* transcription, was significantly more expressed in late-bolting B2557 in the 1L7_vs_1L4 comparison. Meanwhile, we found that the expression level of *BoCO*, which is the upstream promoter of *BoFT*, was higher in the leaves than in the apical flower buds. However, *BoCO* in the leaves was not significantly different between the two broccoli varieties. The photoreceptor *BoPHYA*, which perceives light signals, was more expressed in the apical flower buds than in the leaves. There was also a significant difference in its expression levels between 2B4 and 2B7, with higher expression in the apical flower buds of early-bolting B2554. The expression levels of *BoLWD1*, which is related to biological rhythms, were significantly different in 1B7_vs_1B4, 2B7_vs_2B4, 1L7_vs_1L4, and 2L7_vs_2L4, and its expression in early-bolting B2554 was higher than that in late-bolting B2557.

The expression of *BoFLC*, which is a flowering repressor gene in the vernalization pathway, was significantly different in 1B7_vs_1B4, 2B7_vs_2B4, 1L7_vs_1L4, and 2L7_vs_2L4 and was higher in late-bolting B2557. This result was consistent with the observed phenotype. The expression of *BoVRN1*, which inhibits *BoFLC* expression, was significantly different in 1B7_vs_1B4 and 2B7_vs_2B4 and was higher in early-bolting B2554. The expression level of *BoUBC2*, a positive regulator of *BoFLC*, was significantly different in 1B7_vs_1B4, 2B7_vs_2B4, and 2L7_vs_2L4, but it was higher in early-bolting B2554.

Gibberellin (GA) is an important plant hormone in the process of flower transformation in plants. When the synthesis of endogenous GA is blocked or the signal transduction process of GA is disrupted, the flowering process of plants is impacted [9]. We observed that the expression of *BoGA20ox3*, which encodes the key rate-limiting enzyme in gibberellin synthesis, was much higher in the leaves than in the apical flower buds, and its expression was even higher in early-bolting B2554. The expression of *BoGA2ox2* in the apical flower buds was much higher than that in the leaves, and it was significantly different in 1B7_vs_1B4, with higher expression in early-bolting B2554. Additionally, the GA receptor *GID1* (*GIBBERELLIC INSENSITIVE DWARF 1*) regulates GA signal transduction by binding to DELLA proteins. Three *BoGID1B* were identified in broccoli, and their expression levels in the apical flower buds were higher than in the leaves. Their expression levels were significantly different in 1B7_vs_1B4 and were relatively high in early-bolting B2554. *SVP* (*SHORT VEGETATIVE PHASE*) negatively regulates *FT* and *SOC1* (*SUPPRESSOR OF OVEREXPRESSION OF CO 1*) but is negatively regulated by GA. *BoSVP* expression was much higher in the leaves than in the apical flower buds, but its expression in the apical flower bud tissues was significantly different in 1B7_vs_1B4, with a higher expression level in late-bolting B2557.

*SPLs* (*SQUMOSA PROMOTER BINDING PROTEIN LIKES*) belong to a class of transcription factors that promote flowering in the aging pathway. As the plant age increases, the miR156 levels decrease, and *SPL* expression increases, thus promoting flowering [28]. We observed that *BoSPL3* and *BoSPL5* exhibited higher expression levels in the apical flower buds than in the leaves, both being significantly different in 1B7_vs_1B4 and more highly expressed in early-bolting B2554. In contrast, the expression of *BoSPL4* in the leaves was higher than that in the apical flower buds. Despite its higher expression level in the leaves, its expression pattern was also significantly different in 1B7_vs_1B4 and had higher expression in early-bolting B2554.

### 2.5. Differential Expression Analysis in Phytohormone Signal Transduction Pathways

In the plant hormone signaling pathway, we identified four DAMs related to endogenous plant hormones. ABA and IAA were significantly different in 2B7_vs_2B4, and their concentrations in early-bolting B2554 were higher than in late-bolting B2557, and their accumulation was twice as much as in B2557 (Figure 5). JA and JA-Ile were significantly different in 1B7_vs_1B4, with their concentrations being higher in late-bolting B2557 than in early-bolting B2554, and their accumulation was ten times greater than in B2554 (Figure 6). Additionally, SA was detected, but no significantly differential expression was observed when compared among the various tissues (Figure 6).

We further analyzed the detected plant hormones and found that the biosynthesis and signal transduction of auxin had different activity levels in the two broccoli varieties. We discovered that 23 auxin biosynthesis and signal transduction-related genes were differentially expressed among the two varieties and their studied tissues. Specifically, one of the tryptophan synthases, *BoTSB2-1*, was significantly differentially expressed in the 1B7_vs_1B4 comparison, and *BoTBS2-2* was significantly differentially expressed in 1L7_vs_1L4. The tryptophan aminotransferase *BoTAA1* was significantly differentially expressed in all four comparison groups and showed opposite expression patterns in the different tissues. The *BoTAA1* content in the apical flower buds was higher in B2554 than in B2557, and the *BoTAA1* content in the leaves was higher in B2557 than in B2554. During auxin signal transduction, the overall expression of the *auxin response factor* (*ARF*), *F-box transport inhibitor response 1* (*TIR*), and *Gretchen Hagen 3* (*GH3*) genes in the apical flower buds was higher than that in the leaves. Among them, the three *BoTIR5* genes were significantly differentially expressed in 1B7_vs_1B4 and highly expressed in early-bolting B2554, and *BoGH3.6* was significantly differentially expressed in 1B7_vs_1B4 and 2B7_vs_2B4 and highly expressed in late-bolting B2557. Seven auxin/indole-3-acetic acid (Aux/IAA) transcriptional repressors (*BoIAAs*) and five *BoSAURs* were differentially expressed between the early- and late-bolting broccoli. Their expression levels varied within and between the varieties and across the different growth stages, indicating the role of auxin signaling in the bolting process.

During abscisic acid signal transduction, we observed that the expression levels of *BoPLY4*, *BoPP2C08*, and *BoABF5* were significantly different in 1B7_vs_1B4 and 2B7_vs_2B4. *BoPP2C08* was expressed at a higher level in B2557. The expression patterns of *BoPLY4* and *BoABF5* were different. The expression of *BoPLY4* was higher in B2554 during the first growth period and higher in B2557 during the second growth period. In contrast, the expression of *BoABF5* was higher in B2557 during the first growth period and higher in B2554 during the second growth period. Additionally, there were two *BoPLYs*, five *BoPP2Cs*, and three *BoNCEDs* that were differentially expressed between the early- and late-bolting broccoli.

During the process of jasmonic acid signal transduction, the expression of jasmonic acid early response genes is inhibited by the JAZ protein, which prevents the *MYC2* transcription factor from being active. We found that 10 *JAZ* (*Jasmonate ZIM-domain*) family genes exhibited differential expression levels between the early- and late-bolting broccoli, with most *JAZs* showing higher expression in the apical flower buds and significantly different expression levels in 2B7_vs_2B4, with higher expression in late-bolting B2557. In addition, we identified 11 differentially expressed genes related to jasmonic acid biosynthesis between the early and late growth stages including five *BoLOXs*, one *BoAOC*, one *BoOPR*, two *BoOPCLs*, and one *BoACX*.

Although the expression levels of the detected salicylic acid were not different between the plant organs, we analyzed the related genes involved in salicylic acid biosynthesis and signal transduction. The heterobranching acid synthase ICS, which is an important enzyme in the biosynthesis pathway that catalyzes the conversion of chorismic acid (CA) to isochorismate (ISC), was found to be more highly expressed in the apical flower buds of broccoli, with significant expression differences in the first growth period. *PBS3* (*AVRPPHB SUSCEPTIBLE 3*) is another key enzyme in the plant SA biosynthetic pathway, and we found that three homologous *BoPBS3* were more highly expressed in the leaves. Furthermore, one *BoTGA* gene was differentially expressed in the leaves. Shikimic acid, required for salicylate synthesis, exhibited a higher content in the leaves, and a higher concentration in the B2557 leaves was observed.

### 2.6. Differential Regulation in Sucrose Metabolism and Transport Pathways

Carbohydrates such as sucrose are abundant during the development of flower buds and organs. They regulate plant growth and differentiation by producing sugar signaling molecules via sucrose metabolism and their direct or indirect interactions with other signaling pathways (Figure 7).

Among the differentially expressed genes we discovered, three transcripts related to sugar transporters, including hexose and sucrose transporters SWEET (BoSWEET1, BoSWEET10) and hexose transporter STP4, were significantly differentially expressed in 1B7_vs_1B4 and 2B7_vs_2B4. The transcripts of these three genes were highly expressed in early-bolting B2554. Except for the expression of *BoSWEET1*, which was significantly different in 1L7_vs_1L4, the expression of other genes in the leaves was basically no different, among which *BoSWEET10* was basically not expressed in the leaves. Moreover, the transcripts of the *BoSPS* and *BoINV* genes related to sucrose metabolism showed no significant difference in the leaf comparison group, but was significantly different in the apical flower bud comparison groups 1B7_vs_1B4 and 2B7_vs_2B4, and were also highly expressed in early-bolting B2554. The sucrose transporter *BoSUS1* was significantly different in the apical flower buds of 1B7_vs_1B4, and the overall expression was much higher in the apical flower buds than in the leaves. *TPS1* (*TREHALOSE-6-PHOSPHATE SYNTHASE 1*) is associated with trehalose-6-phosphate biosynthesis. Two *BoTPS1* genes were significantly different in 2B7_vs_2B4 and were highly expressed in late-bolting B2557. *BoTPS8* was significantly different in 1B7_vs_1B4 and was highly expressed in early-bolting B2554. The expression of these three genes in the leaves was basically the same. Sucrose synthesis is accomplished through a series of enzymatic catalysis steps and the regulation of glucose. We observed that sucrose accumulation in early-bolting B2554 was higher than in late-bolting B2557, both in the apical flower buds and leaves, and the content was the highest in the leaves of early varieties during the second growth period. The glucose content was found to be differentially regulated during the second growth stage and highly expressed in early-bolting B2554.

### 2.7. Differential Regulation in Phenylpropanoid and Flavonoid Biosynthesis Pathways

Studies have revealed that most phenolic acids are metabolic products derived from the phenylpropanoid biosynthesis pathway. Moreover, p-coumaroyl-CoA generated in this pathway serves as the starting precursor for flavonoid biosynthesis. Metabolomic profiling revealed distinct accumulation patterns between early- and late-bolting lines (Figure 8). In the 1B7_vs_1B4 and 2B7_vs_2B4 comparisons, eight phenylpropanoid-pathway metabolites—p-coumaric acid, sinapic acid, chlorogenic acid, sinapaldehyde, coniferaldehyde, coniferyl alcohol, coniferin, and eugenol—showed significantly higher accumulation in early-bolting B2554. In contrast, no metabolites with statistically significant content changes were observed in 1L7_vs_1L4. Notably, the accumulation of coumaric quinic acid and pinoresinol showed significant variation in 2L7_vs_2L4, with their contents significantly increasing in late-bolting B2557.

A total of 32 phenylpropanoid and flavonoid biosynthesis-related genes showed differential expression between early- and late-bolting broccoli. Among these, cinnamate-4-hydroxylase gene *C4H*, 4-coumarate-CoA ligase gene *4CL*, caffeic acid O-methyltransferase gene *COMT*, coumarate-3-hydroxylase gene *C3H*, and cinnamyl alcohol dehydrogenase gene *CAD* were significantly different in 1B7_vs_1B4 and 2B7_vs_2B4, and all were highly expressed in early-bolting B2554. This was consistent with the significant increase in the downstream metabolite content catalyzed by these enzymes in early-bolting B2554.

## 3. Discussion

The stem apical meristem of the broccoli has already produced flower buds after bolting. Given the scarcity of genetic data for controlling the bolting time of broccoli, we referred to the flowering time (Ft) gene discovered in the flowering research of *A. thaliana*. Then, we analyzed the changes of these genes in the early- and late-bolting broccoli, and made some reasonable speculations on the early- or late-bolting of broccoli varieties. Flowering time is one of the most important developmental factors in the life cycle of plants. In *A. thaliana*, six genetic pathways, including the photoperiod, vernalization, temperature, gibberellin, autonomous, and age pathways that regulate the flowering time, have been identified. By comparing the known Ft-related genes in these six pathways with homologs in broccoli, we identified 293 Ft genes in the transcriptome of broccoli, among which 39 were differentially expressed between early-bolting B2554 and late-bolting B2557. Therefore, approximately 85% of the Ft genes were expressed at similar levels in the two varieties. We speculated that the 39 Ft-related DEGs could provide some clues for the late bolting of B2557. In addition, several studies have suggested that the flowering time of plants is also influenced by hormone- and sugar-dependent pathways. Therefore, We identified several DEGs and DAMs related to hormones and sugars in our transcriptomes and metabolomes. Analyzing the differences between early-bolting B2554 and late-bolting B2557, we can provide some valuable clues for the mechanism of broccoli bolting.

### 3.1. The Interactions Between Flowering Locus T and Its Promoter/Repressors Regulate Broccoli Bolting

The core component *CONSTANS* (*CO*) in the photoperiodic pathway is an important gene that allows plants to respond to photoperiodic regulation and monitor the length of daylight. It can transform light and circadian rhythm signals into flowering signals, thereby activating the expression of the *FT* gene and inducing plant flowering. The regulation of *CO* occurs in the leaves rather than in the flowers, where it is initially synthesized [30]. We found that *BoCO* is mainly expressed in the leaves of both early-bolting B2554 and late-bolting B2557, and their expression levels were the same in the two varieties. However, the expression levels of *BoFT* in the leaves of the two varieties were different. We speculate that *BoCO* directly activates the expression of *BoFT* in the leaves, while the low expression of *BoFT* in late-bolting B2557 is due to it being inhibited by other repressors. *FT* is an integrator factor that plays a key role in floral induction. Therefore, the low expression of *BoFT* in the apical flower buds of B2557 may be the main factor causing late bolting in this variety [5,27]. *TEM1* (*TEMPRANILLO 1*) and *TOEs* (*TARGET OF EATs*) can bind to specific cis-elements of the *FT* gene to inhibit its transcription [31,32]. These repressors suppress the expression of *FT*, preventing plants from bolting. *BoTOE1-1* and *BoTOE1-2* in B2557 had higher expression levels in the apical flower buds than in B2554, which may be the reason behind the low expression of *BoFT* in B2557. Another gene, *BoTEM1,* only showed this phenomenon during the second growth stage. However, *BoTOE2* exhibited a different expression pattern. Compared with B2557, *BoTOE2* was more highly expressed in B2554, and its concentration gradually increased in both varieties over time. Based on the fact that the expression levels of these floral repressors in the miR172/TOE1-TOE2 signaling cascade need to drop below a critical threshold to trigger flowering, it has not yet been confirmed as to whether *TOE1* and *TOE2* are downregulated at the same time [33]. We speculate that *BoTOE2* may function at other times or may have undergone functional differentiation during evolution. In addition, *BoLWD1* showed expression level differences in the apical flower buds and leaves of both early- and late-bolting varieties. It may potentially be an important signaling molecule that regulates bolting, given that the *LWD1* (*LIGHT-REGULATED WD1*) protein mainly functions near the biological clock that controls photoperiodic flowering [34].

We found that *FLC* was associated with the bolting phenotypes of the two broccoli varieties via the vernalization pathway. The expression trends of *BoFLC* and *BoVRN1* indicate that *FT* and *SOC1* may be inhibited in B2557. This agrees with the fact that *FLC* inhibits *FT* in *A. thaliana* and is consistent with the downregulation of the flower-inhibiting factor *FLC* by *VRN1* (*VERNALIZATION1*) [35,36]. The concentration of another flowering inhibitor *BoSVP* in the leaves remained basically unchanged, although its expression levels in the leaves were higher than in the apical flower buds. However, the expression of *BoSVP* in the apical flower buds of late-bolting B2557 was higher than that in early-bolting B2554, indicating that it may be inhibiting *FT* expression in B2557. Since *SVPs* inhibit the expression of genes that initiate flowering by forming complexes with *FLC*, they play a role in delayed flowering [37]. Therefore, it can be speculated that flowering repressors, such as *FLC* and *SVP*, jointly inhibit the early bolting of broccoli by suppressing the expression of *FT*.

On the other hand, gibberellin was not detected in either of the two materials. However, the high expression of *BoGA20ox*, the rate-limiting enzyme for gibberellin synthesis, in the leaf tissue of early-bolting B2554 may have promoted the regulation of GA biosynthesis [38]. GA can promote the expression of *FT* in plants under long day light [39], so the high expression of *BoFT* in the leaves may be affected by the expression of *BoGA20ox*. Meanwhile, during the first period, the high expression of gibberellin receptor *BoGID1B* in the apical flower bud of early-bolting B2554 is consistent with observations in *A. thaliana* [40]. *GID1* (*GIBBERELLIN INSENSITIVE DWARF1*) binds to DELLA to accelerate its degradation, while the flower-promoting factors *SOC1* and *LFY* (*LEAFY*) are both down-stream targets of DELLA proteins in *A. thaliana* [10]. Additionally, the gibberellin deactivating enzyme *BoGA2oxs*, which is highly expressed in the apical flower bud of early-bolting B2554 but less so in all leaves, may be an important factor maintaining GA turnover in the development of the apical flower bud tissues [41]. Among these observations, it is speculated that the changes in the expression of the GA pathway may promote broccoli bolting in early growth.

### 3.2. Activated Flower Meristem Genes Regulate the Bolting of the Apical Flower Buds in Broccoli

FT forms a dimer with FD and activates downstream *SOC1* and multiple meristem genes. During the first growth period, *BoFT*, *BoFD*, and *BoSOC1* exhibited consistent expression patterns in B2554 compared with B2557, promoting early bolting in B2554. Some meristem genes were mostly expressed at higher levels in the apical flower buds, while they were low or not expressed in the leaves. The activated downstream genes *LFY*, *FUL* (*FRUITFULL*), and *AP1* (*APETALA1*) collectively determine the formation of floral organs [3]. Therefore, the high expression of *BoAP1*, *BoFUL*, and *BoLFY* in B2554 may promote early bolting. In contrast, the expression of *BoAGL24* and *BoSEPs* in B2557 was much lower than that in B2554, which limited bolting in B2557. Since *AGL24* (*AGAMOUS-LIKE 24*) and *SEP3* (*SEPALATA3*) are important factors for identifying the flower meristem, it is speculated that low expression levels of these genes contributed to late bolting in B2557 [42]. The *TFL1* gene, which is an inhibitor of *LFY* and *AP1*, is essential for transitioning from inflorescence meristems to floral meristems [3]. The low expression of meristem identity genes, such as *BoAP1* and *BoLFY*, in B2557 was consistent with the high expression of the *BoTFL1* gene. Their low expression may be related to the high expression of *BoTFL1*.

### 3.3. The Relationship Between Hormone Levels and the Early- and Late-Bolting Phenotypes

The auxin signaling pathway plays a crucial role in determining flowering fate such as regulating the initiation and growth of floral meristems and the reproductive success of flowers [8]. The expression of genes involved in auxin biosynthesis in the studied broccoli tissues varied with increasing growth time, and different tissues also exhibited tissue-specific auxin levels. We observed that the expression of the auxin synthesis gene *BoTAA1* was the highest in the apical flower bud tissues of B2445 during the second growth stage. Meanwhile, a large amount of auxin was observed in the apical buds of early-bolting B2554. *Auxin response factors* (*ARFs*) play a role in signal transduction by binding to the cis-acting elements of auxin early response genes to regulate their transcriptional expression. We found a higher accumulation of *BoARF5* in the apical flower buds of early-bolting B2554, and the early response genes *BoAUX1*, *BoSAU20*, and *BoSAU12* involved in auxin-mediated responses were also more highly expressed in these plants. Therefore, it was speculated that auxin may play an important role in regulating early bolting [8]. At the same time, the expression of *BoARF* in the apical flower buds may be associated with that of *BoLFY*, promoting bolting in B2554 during the early growth stage. This is based on the fact that auxin/ARF can trigger the activation of floral primordia by enhancing the expression of *LFY* [43]. Additionally, we observed that the expression levels of certain auxin-related genes, including *GH3*, *SAUR*, and *AUX/IAA*, were significantly different among the tissues and growth stages between B2554 and B2557. For instance, *BoIAA3*, *BoIAA17*, *BoIAA2*, *BoGH3.6*, *BoSAUR78*, and *BoSAUR3* were expressed at higher levels in the apical flower buds of B2557. Members of these gene families have undergone gene duplication events during evolution, which may have resulted in genetic redundancy, so they may have overlapping functions or exhibit unique localized functions [44,45]. For these reasons, it was speculated that these genes may be the underlying factors driving the delayed growth of late-bolting broccoli.

During the initial stage of bolting, there were no significant differences in the ABA content in the apical flower buds. During the second growth stage, the ABA content in B2554 significantly increased, which may be due to the higher expression of *BoNCED* in B2554 than in B2557 during this growth period. *NCEDs* are the main regulators of ABA levels [46] and may play a significant role in inducing bolting in B2554. JAZ proteins are transcriptional repressors in the jasmonic acid signaling pathway and can counteract the inhibition of *FT* by the flowering suppressor *TOE* [47]. Our research showed that the expression of most JAZ transcriptional repressors was relatively high and similar in the apical flower buds from the first growth period, but *BoTOE1* was expressed at a higher level in B2557 than in B2554. At the same time, the concentration of JA in B2557 was much higher than in B2554, so we speculated that *BoTOE1* is one of the main factors that causes B2557 to bolt late. The high JA content in B2557 can induce the ubiquitination of the JZA protein that deinhibits *BoTOE1*, resulting in the lower expression of *FT* and, consequently, late bolting [47]. In addition, certain genes related to JA synthesis in the JA biosynthesis pathway were differentially expressed in different tissues, which may also affect the bolting time in broccoli. Furthermore, we found that there were no significant differences in the expression levels of salicylic acid in the salicylic acid signal transduction pathway in the various tissues, but the expression of *BoPBS3* was consistent with that of salicylic acid in the leaves. *PBS3* is one of the key enzymes involved in salicylic acid synthesis and promotes salicylic acid synthesis [48]. Salicylic acid can affect the expression of *FT* and plays a role in flowering, so it was speculated that *BoPBS3* may have promoted bolting [49]. *BoTAG1* in the leaves was differentially expressed in early-bolting B2554, which was speculated to be one of the potential factors promoting bolting.

### 3.4. Sugar-Exporting Transporters Likely Regulate the Early Bolting of the Apical Flower Buds of Broccoli and Play a Crucial Role in the Long-Distance Transportation Process of Sugar

Our research indicates that alterations in the biosynthesis and transport of sucrose may induce an earlier or later bolting time in broccoli. Since sucrose phosphatase synthase (SPS) is the main rate-limiting enzyme in sucrose synthesis, it determines the rate at which sucrose accumulates [50]. In the transcriptomes, we observed that *BoSPS* was more highly expressed in B2554, so sucrose was more abundant in early-bolting B2554. Sucrose acts as a signaling molecule that primarily, but indirectly, promotes the expression of *FT* in the leaf phloem. Moreover, sucrose is transported from source to sink via different sugar transporters (such as SUC and SWEET) or through plasmodesmata (PD) connected to the SE/CC complex [22]. SWEET, which is a novel sugar transporter, mediates the passive diffusion of sucrose across the cell membrane along the concentration gradient. We observed that *BoSWEET10* was mainly expressed in the apical flower buds and not in the leaves, while *BoSWEEET1* was expressed higher in B2554 during the first period, both of which were consistent with the expression pattern of sucrose. Metabolic data confirmed that sucrose accumulation was higher in B2554, allowing us to identify a correlation among the expression of *SWEET*, sucrose levels, and *FT* production [51,52]. The sugar signaling molecules produced by the decomposition of sucrose can activate or inhibit the expression of certain genes related to flowering. For instance, sucrose can be detected by *TPS* to induce T6P synthesis, thereby indirectly promoting the expression of *SPL* by inhibiting the expression of miR156 and initiating the flowering process [16,28]. The synthesis of T6P is regulated by the *TPS* gene and serves as a flowering-inducing signal in *A. thaliana*. Although there was no difference in the T6P content between the apical flower buds and leaves, *BoTPS1* and *BoTPS8* showed opposite expression patterns in the apical flower buds, which may be required for flower bud differentiation at this time. In addition, the expression levels of *BoSUS1*, *BoINV4*, and *BoSTP4* were different in the early- and late-bolting tissues, which could also serve as potential signals to regulate the bolting time.

### 3.5. Phenolic Compounds May Affect the Early Bolting of Broccoli

Studies have shown that phenolic compounds may affect flower bud differentiation in plants, while flavonoids are essential for pollen germination and pollen tube growth in some crops such as maize and petunia [53,54]. The oleuropein content was observed to drop sharply when the olive transitioned from the vegetative stage to the reproductive stage [55]. The low-dose exogenous application of oleuropein was found to inhibit flowering in *Kalanchoe blossfeldiana* [56]. This indicates the negative impact of oleuropein on flower induction. The accumulation of phenolic compounds, such as cinnamic acid and chlorogenic acid, may contribute to low-temperature-induced flowering in *Pharbitis nil* [57]. Conversely, the significantly reduced levels of ferulic acid, caffeic acid, and benzoic acid may promote flower bud differentiation in *Mikania micrantha* at a 900-m altitude [58]. In this study, the levels of *p*-coumaric acid, sinapic acid, chlorogenic acid, sinapaldehyde, caffealdehyde, coniferyl alcohol, coniferin, and eugenol were significantly increased in early-bolting B2554, suggesting that these compounds may play a significant role in promoting bolting. However, whether these compounds directly affect bolting still requires further research. In addition, key enzymes in the phenylpropanoid biosynthesis pathway, such as C4H, 4CL, COMT, and C3H, were found to play a crucial role in phenolic acid biosynthesis [59,60]. In this study, the genes encoding these enzymes showed a significant difference between early-bolting B2554 and late-bolting B2557, including *BoCH4-4*, *Bo4CL4-1*, *Bo4CL4-2*, *Bo4COMT-2*, *BoCOMT-3*, and *BoC3H*, and were highly expressed in early-bolting B2554. This is consistent with the result that the corresponding metabolite contents were significantly increased in early-bolting B2554. Therefore, it is speculated that these genes positively regulate phenolic acid biosynthesis in the phenylpropanoid biosynthesis pathway of broccoli.

## 4. Materials and Methods

### 4.1. Plant Materials

B2554 is an early-bolting inbred broccoli line, and B2557 is a late-bolting inbred line. The apical flower bud tissues and leaf tissues of the early- and late-bolting plants were sampled at different growth stages. When the B2554 plants began to bolt (i.e., when the apical flower bud tissue was 1 cm), it was considered as the first growth period (Figure 9A). The apical flower bud tissues (abbreviated as B) and leaf tissues (abbreviated as L) of B2554 and B2557 were sampled and named 1B4, 1B7, 1L4, and 1L7, respectively. Then, when the two plants had grown for sixteen days, they were considered to be in the second growth period. For the sampling of the apical flower bud meristem, we cut the green part of the flower ball and then cut the middle of the bulb to take the growth point position. At the same time, the sampling site of the leaf was the same as that of the first period, and the samples were named 2B4, 2B7, 2L4 and 2L7, respectively. All plant materials were handled in the field following standard agricultural practices, with three biological replicates collected per sample. All samples were flash-frozen in liquid nitrogen and stored at −80 °C. All plants were provided by the Institute of Vegetables and Flowers, Chinese Academy of Agricultural Sciences, and were planted at the Shunyi Base of the Institute of Vegetables and Flowers, Chinese Academy of Agricultural Sciences, on 6 August 2023.

### 4.2. Metabolomics Analysis

The apical flower buds (1B4, 1B7, 2B4, and 2B7) and leaves (1L4, 1L7, 2L4, and 2L7) of broccoli were freeze-dried and ground. The subsequent sample preparation and extraction procedures for the broad-target metabolomics analysis followed the method of Zou et al. [61]. The filtered solution was then stored in injection vials for subsequent UPLC-MS/MS analysis. The compounds were analyzed using an UPLC-ESI-MS/MS system (UPLC, ExionLC™ AD, https://sciex.com.cn/) and tandem mass spectrometry system (https://sciex.com.cn/). The liquid chromatography, solvent system, and ESI source operation were also conducted in accordance with the method of Zou et al. [61]. In the metabolomics analysis, principal component analysis (PCA) was conducted using the statistical function prcomp in R (www.r-project.org). Hierarchical clustering analysis (HCA) and Pearson correlation coefficients (PCC) was performed using the Metware Cloud, a free online platform for data analysis (https://cloud.metware.cn). In the comparative analysis of the two broccoli varieties, the variables of interest (VIP > 1) and *t*-test *p*-values (*p* ≤ 0.05) determined the differentially accumulated metabolites (DAMs). Finally, the identified metabolites were annotated using the KEGG database (http://www.kegg.jp/kegg/compound/ (accessed on 1 December 2023)).

### 4.3. Transcriptome Analysis

The apical flower buds (1B4, 1B7, 2B4, and 2B7) and leaves (1L4, 1L7, 2L4, and 2L7) of broccoli were collected for cDNA library construction and transcriptome sequencing. The cDNA library construction and sequencing were performed by Metware Biotechnology Co., Ltd. in Wuhan, China. The RNA was extracted from eight tissues (1B4, 1B7, 2B4, 2B7, 1L4, 1L7, 2L4, and 2L7) using the CTAB-PBIOZOL method [62]. Subsequently, cDNA libraries were constructed by fragmenting the enriched mRNA into small fragments under appropriate conditions and synthesizing cDNA using hexamer primers. The adapters were then added to both ends of the cDNA fragments, and the libraries were purified. Each library was quantified by qPCR using a Qubit instrument (effective concentration > 2 nM). Finally, the libraries that had been qualified were sequenced on the Illumina HiSeq platform. The low-quality reads were removed, and only the high-quality clean reads were retained. The distribution of sequencing error rates and GC content of the data were then examined [63]. The clean reads were mapped to the reference genome of broccoli [64]. A routine bioinformatics analysis was conducted using bioinformatics tools and R v3.5.1 packages. Based on the adjusted FPKM, differential expression analysis between different tissue samples from the same growth period was performed using DESeq2 v1.22.1 [65]. The genes with |log2 fold change| ≥ 1 and FDR < 0.05 were considered as DEGs, and GO and KEGG enrichment analyses were conducted for the DEGs of each comparison group [66,67]. The enrichment results were visualized in scatter plots.

### 4.4. qRT-PCR Validation of Gene Expression

We performed qRT-PCR expression analysis of nine genes to verify the gene expression results obtained from the RNA sequencing. The primers were designed using the primer-blast program (http://brassicadb.cn/#/BLAST/ (accessed on 5 January 2024)), and the primer sequences for each gene are listed in Appendix A. qRT-PCR was performed using the CXF96 Touch instrument (Bio-Rad, Hercules, CA, USA) and Taq Pro Universal SYBR qPCR Master Mix (Vazyme, Nanjing, China). Three biological replicates were included for each tissue sample. The reference gene was β-actin, and the relative expression level was calculated using 2^−ΔΔCT^.

## 5. Conclusions

This study analyzed the transcriptome and metabolome profiles of B2554 and B2557 to understand broccoli bolting. The upregulation of genes, such as *FT*, *LFY*, *AP1*, *FUL*, and *SEPs*, and the downregulation of genes, such as *FLC*, *TFL1*, and *TOE1*, in B2554 were found to play a possibly significant role in promoting bolting. The upregulation of the sugar transporter SWEET in B2554 contributed to the accumulation of sucrose, and this sucrose accumulation may induce bolting. In the apical flower buds of the first growth period, auxin response factors *Aux1* and *ARF* were upregulated in B2554, which may play an important role in regulating early reproductive transition. In addition, some genes in the ABA, JA, and SA pathways may interact with certain flowering genes in the flowering pathway to jointly affect bolting in broccoli. Phenolic acids may affect the bolting time of broccoli. In summary, together with the flower-related pathways, the genes involved in the biosynthesis and signal transduction of hormones (IAA, JA, ABA, and SA) and the biosynthesis pathway of sugar may jointly regulate the observed phenotypes. This study can provide a valuable reference for the subsequent analysis of molecular regulatory mechanisms regulating broccoli bolting time.

## Figures and Tables

**Figure 1 ijms-26-03726-f001:**
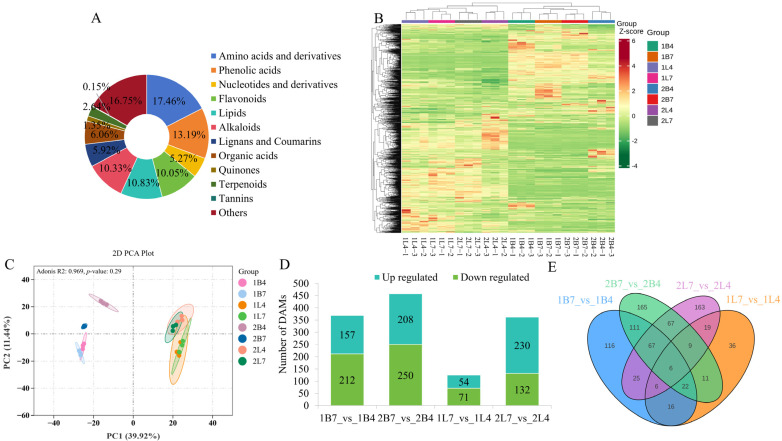
Summary of the metabolome. (**A**) Component analysis of the identified metabolites from apical flower buds and leaves. (**B**) Heatmap of all metabolite abundances. (**C**) Principal component analysis. (**D**) Upregulated and downregulated differentially accumulated metabolites (DAMs) in each differential comparison group. (**E**) Venn diagram of the differential expression patterns of the four groups.

**Figure 2 ijms-26-03726-f002:**
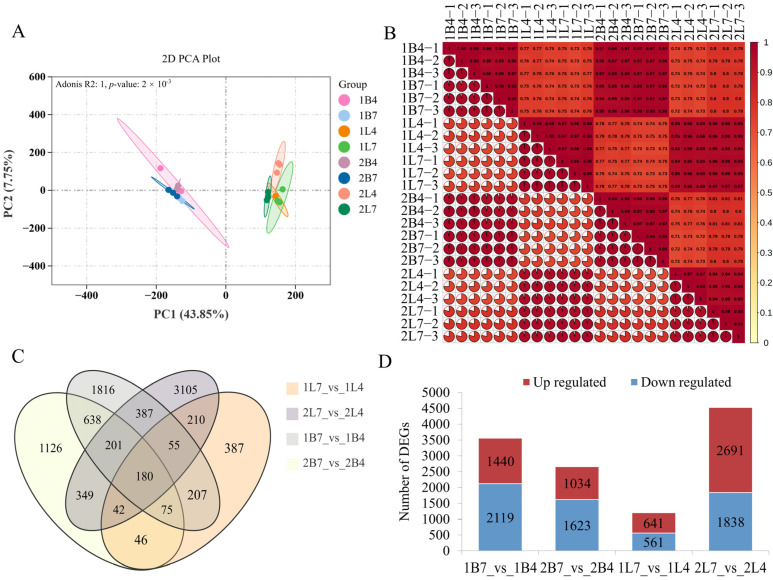
Summary of the transcriptome. (**A**) Principal component analysis. (**B**) Sample correlation heatmap. (**C**) Venn diagram of differentially expressed genes (DEGs) in each differential comparison group. (**D**) Upregulated and downregulated DEGs in each differential comparison group.

**Figure 3 ijms-26-03726-f003:**
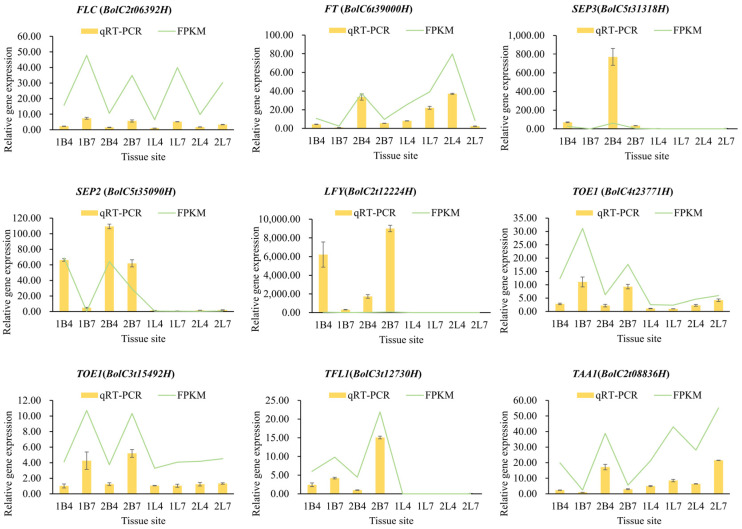
qRT-PCR results from nine flowering-related genes from 1B4, 1B7, 2B4, 2B7, 1L4, 1L7, 2L4, and 2L7. B represents the apical flower bud, L represents the leaf; 1 represents the first growth period, 2 represents the second growth period. The error bars indicate the standard deviations. The reference gene was β-actin.

**Figure 4 ijms-26-03726-f004:**
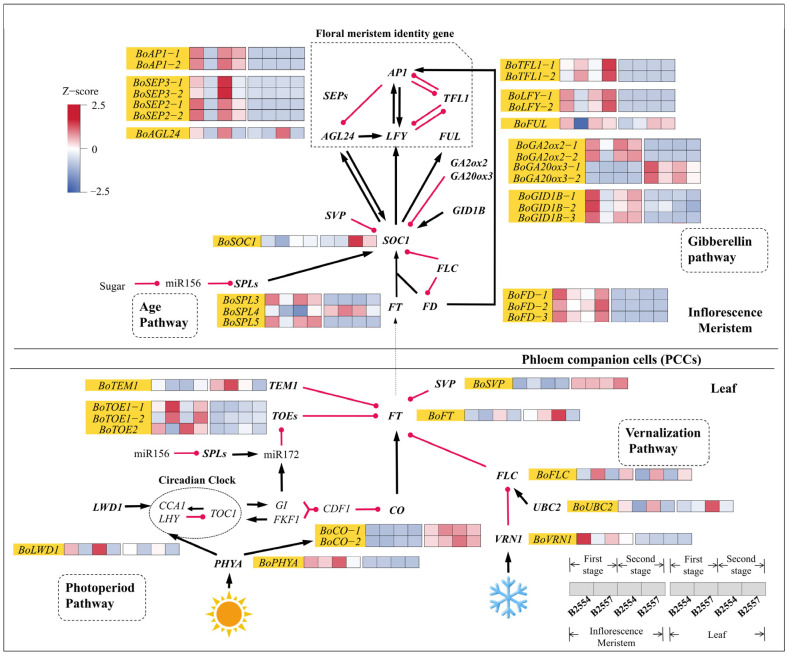
Flowering pathway in broccoli. Heatmaps represent FPKM values for the major flowering-related genes after Z-score transformation. Black arrows indicate promotion, whereas red dots indicate inhibition. The expression patterns of these genes are shown in Appendix A.

**Figure 5 ijms-26-03726-f005:**
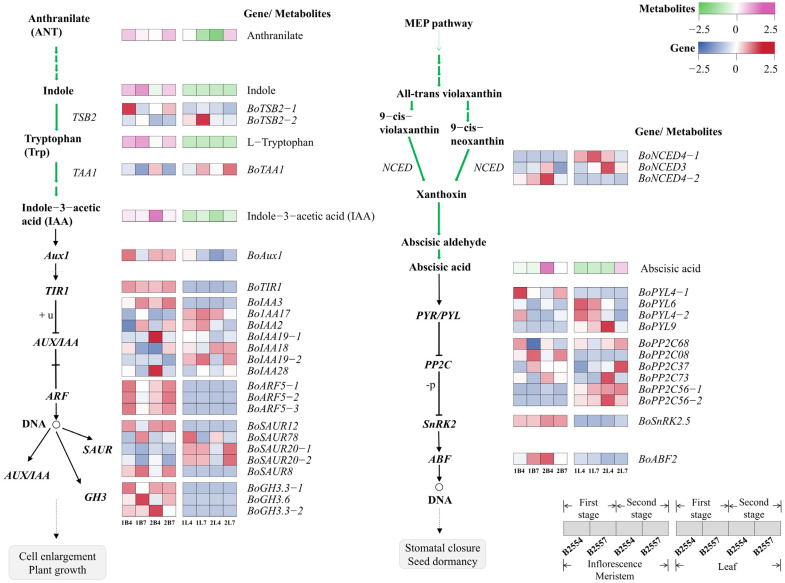
Auxin synthesis and transport (**left**) and the abscisic acid synthesis and transport (**right**) pathways. The heatmap presents the FPKM values after Z-score transformation. The synthetic pathway is indicated by green arrows, and the transduction pathway is indicated by black arrows. Black arrows indicate promotion, and black blunt lines indicate inhibition. The expression levels of these genes are listed in Appendix A.

**Figure 6 ijms-26-03726-f006:**
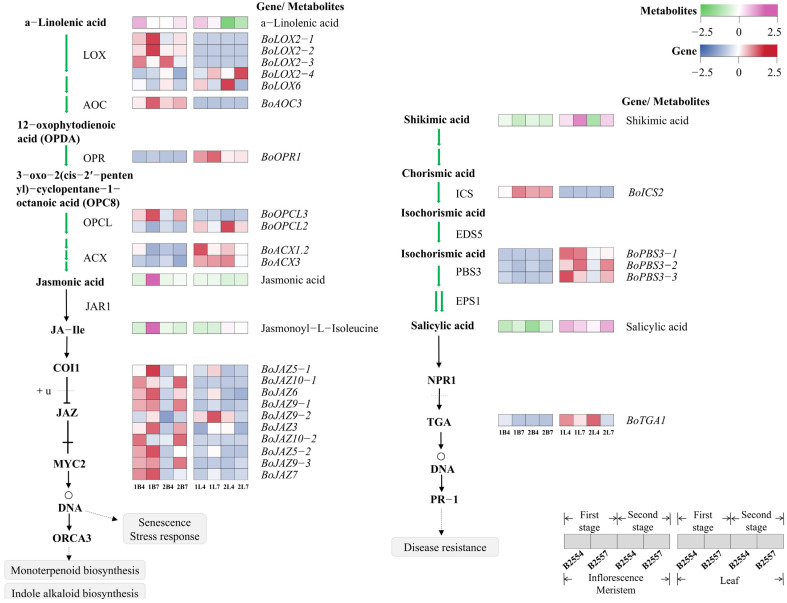
Jasmonic acid synthesis and transport (**left**) and salicylic acid synthesis and transport (**right**) pathways. The heatmap presents the FPKM values after Z-score transformation. The synthetic pathway is indicated by green arrows, and the transduction pathway is indicated by black arrows. Black arrows indicate promotion, and black blunt lines indicate inhibition. The expression patterns of these genes are listed in Appendix A.

**Figure 7 ijms-26-03726-f007:**
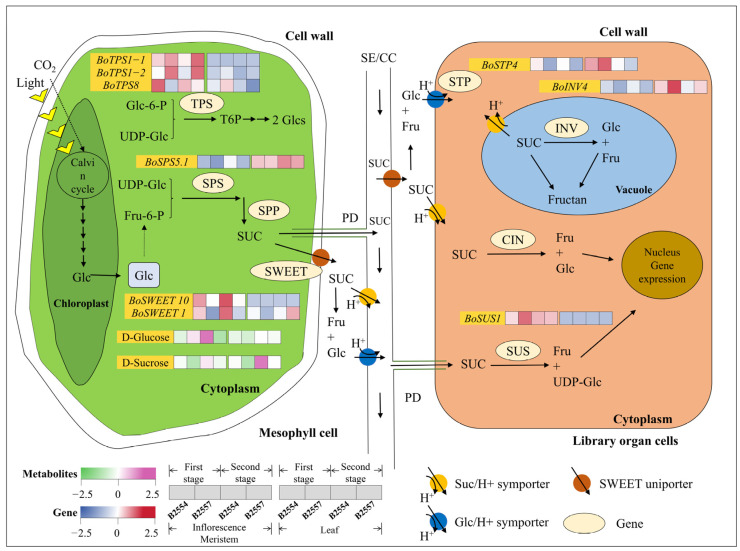
Sucrose synthesis, metabolism, and transport pathway. The heatmap presents the FPKM values for sucrose-associated genes after Z-score transformation. The sucrose-related pathway diagram was modified according to the study of Ruan [29]. The expression levels of these genes are shown in Appendix A.

**Figure 8 ijms-26-03726-f008:**
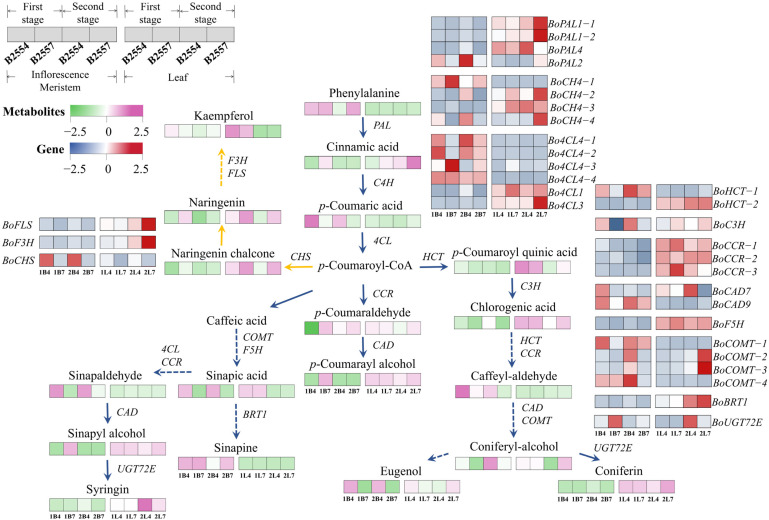
Phenylpropanoid and flavonoid biosynthesis pathways. The blue arrow represents the phenylpropanoid biosynthesis pathway, and the yellow arrow represents the flavonoid biosynthesis pathway. The heatmap presents the FPKM values after Z-score transformation. The expression levels of these genes are shown in Appendix A.

**Figure 9 ijms-26-03726-f009:**
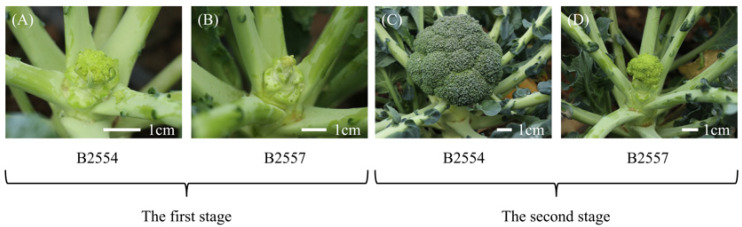
Morphological photographs of the apical flower bud tissues of broccoli. The growth status of the apical flower bud tissues in early- and late-bolting plants during the first and second growth stages. (**A**) 1B4, (**B**) 1B7, (**C**) 2B4, and (**D**) 2B7.

## Data Availability

All data supporting the findings of this study are available in the paper and its Appendix A published online. The RNA-Seq data have been submitted to Sequence Read Archive with the accession number PRJNA1211282.

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
