# Peer review of "Transcriptome and Metabolome Analyses Offer New Insights into Bolting Time Regulation in Broccoli"

_ijms, 2025, doi:10.3390/ijms26083726_

Round 1
Reviewer 1 Report
Comments and Suggestions for Authors
In this study, authors analyzed the transcriptome and metabolome profiles of broccoli B2554 and B2557, they found that the upregulation of genes, such as FT, LFY, AP1, FUL, and SEPs, and the downregulation of genes, such as FLC, TFL1, and TOE1, in B2554 were found to play a possibly significant role in promoting bolting. In addition, some genes in the ABA, JA and SA pathways may interact with certain flowering genes in the flowering pathway to affect broccoli bolting. The methods and results are acceptable. There are some essential problems should be addressed by authors, which are listed below.
- In the Introduction section, the authors only discussed the effects of plant hormones and carbohydrates on flowering. It is recommended that the authors expand the discussion to include transcriptomics and gene regulation. For example, authors could introduce which differentially expressed genes in other plant species contribute to flowering time variation among different cultivars, along with relevant mechanisms.
- In the Materials and Methods section lacks details on environmental conditions, such as temperature, photoperiod (light duration and intensity), and soil fertility.
- In Section 2.1, the authors state: “In the comparison between the apical flower buds and the leaves in the first period, the most different metabolites were flavonoids, while in the second period, the most different metabolites were phenolic acids.” However, the subsequent analysis does not discuss how flavonoids and phenolic acids may influence apical bud development or bolting time.
- Most of the discussions related to the metabolome have focused on plant hormones and sugar metabolism. It is recommended that the authors add a new section to the discussion to briefly discuss the effects of some other metabolites on plant flowering and bolting.
Reviewer 2 Report
Comments and Suggestions for Authors
The reviewed manuscript, titled "Transcriptome and metabolome analyses offer new insights into bolting time regulation in broccoli” by Kuang and colleagues is a thorough, comprehensive work that combines an impressive in-depth analysis of the metabolome (by UPLC-MS/MS) and transcriptome by RNA-seq. This alone was quite impressive and yields a wealth of data that would be of interest to a variety of readers! This work was clearly analyzed and presented, and their follow up analysis focused on the flowering genes and their expression during development. All in all, this is a very solid and strong submission, and I would like to take a moment to acknowledge the wonderful job that the authors did on this work!
I do have a few comments, however they are all minor in nature:
Figure 1: There is a wealth of data in this figure, however the formatting as submitted is too small. Please convert this figure to enlarge it in the final submission. I think that this figure could be an entire page in its size. Parts B&C should be significantly larger to all of the reader to read the legends and the labels.
Figure 2: Same comment as above, specifically for part B.
Figure 3: include details on how the relative expression was determined in the figure legend (e.g. Relative expression to actin, etc).
Section 4.3: please deposit the raw data into an appropriate repository, e.g. the SRA, etc. for researchers who are interested in exploring this in more detail.
Round 2
Reviewer 1 Report
Comments and Suggestions for Authors
Good work of the revision